# The Expansion Cracks of Dolomitic Aggregates Cured in TMAH Solution Caused by Alkali–Carbonate Reaction

**DOI:** 10.3390/ma12081228

**Published:** 2019-04-15

**Authors:** Xiaoxiao Chen, Bin Yang, Zhongyang Mao, Min Deng

**Affiliations:** College of Materials Science and Engineering, Nanjing Tech University, Nanjing 210009, China; 1341865250@njtech.edu.cn (X.C.); 825624014@njtech.edu.cn (B.Y.); mzy@njtech.edu.cn (Z.M.)

**Keywords:** alkali–carbonate reaction, expansion, crack, TMAH

## Abstract

In this study, concrete microbars and rock prisms made of dolomitic aggregates were cured in a 1-mol/L tetramethylammonium hydroxide (TMAH) solution at 80 °C to avoid the effect of alkali–silica reaction (ASR) on expansion. The expansion of specimens was only caused by the alkali–carbonate reaction (ACR). The reason that self-made cement was used in this work was to ensure that the Mg^2+^ contained in the brucite originated only from dolomite. Expansion of concrete microbars and rock prisms was measured, the expansion cracks were systematically observed by orthogonal polarizing microscopy, and the products of ACR were analyzed by scanning electron microscopy (SEM) and energy dispersive spectrometry (EDS). The results showed that the dolomite crystals in the dolomitic aggregates reacted with the TMAH solution and resulted in ACR, which formed calcite and brucite and led to cracking of the specimens. The source of the expansion was the dolomite crystals of the dolomite enrichment area. Expansion cracks either extended inside the rock or into the cement phase and eventually disappeared. The alkali–carbonate reaction significantly contributed to the expansion of dolomitic aggregates cured in TMAH solution at a later curing age.

## 1. Introduction

Concrete is widely used in building construction and chemical engineering. The alkali–aggregate reaction is one of the most important factors affecting the durability of concrete. This reaction can be divided into the alkali–silica reaction (ASR) and alkali–carbonate reaction (ACR) according to the active mineral species in the aggregates. ACR means the dolomite in the dolomitic aggregate can react with OH^−^ to form brucite, calcite, and CO_3_^2−^. The reaction equation is shown in Formula (1), where the formed CO_3_^2−^ reacts with the cement hydration product Ca(OH)_2_ to form calcite and OH^−^. The reaction equation is shown in Formula (2). The formed OH^−^ will proceed to the reaction of Formula (1) until the dolomite reaction in the concrete aggregate is used up. Formula (1) is a process in which the volume of the solid phase is reduced by 4.4% [1], and Formula (2) is a process in which the volume of the solid phase is increased by 10.2% [2]:CaMg(CO_3_)_2_ + 2OH^−^ = Mg(OH)_2_ + CaCO_3_ + CO_3_^2−^(1)

CO_3_^2−^ + Ca(OH)_2_ = CaCO_3_ + 2OH^−^.(2)

The expansion mechanism of ACR can be divided into three categories. Firstly, Pagano and Candy [3] investigated the expansion characteristics of alkali-reactive carbonate aggregates with LiCO_3_ and FeCl_3_ and concluded that the expansion of ACR can be ascribed to osmotic swell. Secondly, according to Tong and Tang [4], brucite crystals formed in the process of ACR grow up in confined spaces, bringing about the expansion of carbonate aggregates and crystal pressure expansion. Thirdly, Katayama [5,6], using polarizing microscopy, observed that dedolomitization produced a myrmekitic texture, composed of spotted brucite and calcite within the reaction rim, along with a carbonate halo of calcite in the surrounding cement paste. Therefore, the expansion of ACR results from the alkali–silica reaction and ACR should be classified as ASR. López-Buendia et al. [6] found that ACR in nonsilica-content carbonates was produced by the instability of the carbonate minerals, causing induced dedolomitization in the concrete.

Since Swenson [7] in the 1950s discovered the first example of ACR-damaged concrete engineering in Canada, the expansion reactivity of ACR has attracted the attention of many researchers, especially regarding ACR being the same as ASR. Grattan-Bellew and co-workers [8,9] found a gel in concrete affected by ACR, which was different from gel composed of calcium magnesium aluminum silicate hydrate formed in typical ASR in concrete. Fecteau and Fournier [10] also found ASR gel in the cracks of Kingston rock in Canada. It is believed that the occurrence of the alkali–carbonate reaction provides a channel for the OH^−^ into the interior of the rock aggregate, leading the alkali–silica reaction of the microcrystalline quartz to cause concrete deterioration. Prinčič et al. [11] insisted that dedolomitization is only a dissolution–precipitation process which leaves no expansion. Some scholars have shown that at room temperature, the alkali in an alkaline solution can reduce the dissolution rate of dolomite [12]. Actually, dedolomitization can occur in any place where expansion occurs [13]. However, according to Katayama [14], ACR should be classified as ASR. The above results indicate that the alkali–silica reaction occurs in carbonate rock aggregates containing microcrystalline quartz and the gel contributes to expansion, although the expansion contribution of ACR cannot be ruled out. It is hard to be convinced of the conclusion that ACR and ASR are the same. Mather [15] observed the presence of a silica reaction ring at the interface between an aggregate and cement paste, which is believed to be the main cause of expansion. In addition, Duke [16] believes that the alkali–carbonate reaction exists only at the interface between a concrete aggregate and cement paste. Milanesi and co-workers [13,17,18] used high-porosity dolomite rock in South America to prepare a mortar for rock activity detection. The ACR products of brucite and calcite were detected, but no ASR gel was found in the crack of the tested samples.

Chen et al. [19] found that tetramethylammonium hydroxide (TMAH) [20] can react with dolomite in carbonate rock and expand, but it does not react with ASR active components (such as microcrystalline quartz). According to Thong and Choi [21], TMAH has been widely used in silicon anisotropic etching due to reliable etching of silicon in the preparation of microelectromechanical systems. In this process, TMAH is used as a reactant and SiO_2_ is the reaction product. It can be concluded that TMAH does not react with SiO_2_ because the reactant cannot react with the product. Therefore, TMAH (organic alkali) solution can be an effective way to distinguish the expansion between ASR and ACR in dolomitic rocks. It is necessary to investigate the crack characteristics and microstructure of dolomitic aggregates cured in TMAH solution.

In this work, microbars prepared by dolomitic rocks and self-made cement without K^+^, Na^+^, and Mg^2+^ as well as rock prisms were fabricated to further prove that the alkali–carbonate reaction causes expansion. TMAH solution was utilized as a curing solution to investigate the expansion characteristics caused only by ACR. The characteristics of the cracks resulting from ACR were systematically investigated. In addition, rod-like brucite formed in the process of ACR was found according to scanning electron microscopy and energy dispersive spectrometry (SEM-EDS) analysis.

## 2. Materials and Methods

### 2.1. Materials

Dolomitic rocks from Guizhou province were used in this study. Figure 1 shows the XRD pattern of the different rocks. It can be seen that these rocks were mainly composed of dolomite, calcite, and quartz. Table 1 shows the chemical composition of the dolomitic rocks used in the study. Figure 2 shows the distribution of dolomite crystals inside YMS1 and CG1 rocks. The arrows point to dolomite crystals. From Figure 2a, it can be seen that the dolomite crystals contained in YMS1 rock were dispersive. On the contrary, from Figure 2b, dolomite crystals contained in CG1 rock were mosaic. The distribution of dolomite was obviously different between these rocks. The grain size of the dolomite crystals contained in these rocks was similar.

According to RILEM AAR-5 [22] and RILEM AAR-2 [23] standards, rocks JT1, YMS1, and CG1 possessed both ASR and ACR activities. Rocks CX1, CX2, and SJW2 possessed ACR activity and potential ASR activity, and rock SJW1, with ACR but no ASR activity, was selected. The self-made cement clinker without K^+^, Na^+^, and Mg^2+^ was prepared by calcining analytic reagents (Xilong Science Co., Ltd., Shantou, China), followed by grinding into powders with less than 10% sieve residue. The analytic reagents were directly heated to 1450 °C with a heating rate of 10 °C /min and a dwelling time of 1 h at 1450 °C. Table 2 shows the raw material composition of the cement clinker without alkali. The reason for the cement without alkali was to ensure that the Mg2+ contained in the brucite originated only from dolomite. To further characterize the cement clinker, Rietveld analysis was utilized to investigate the mineral contents of the cement clinker without K^+^, Na^+^, and Mg^2+^, as shown in Table 3. Figure 3 shows the XRD pattern of the self-made cement clinker. Calcium sulfate dihydrate (Xilong Science Co., Ltd.) and clinker powders at a weight of 5/95 were blended for 12 h to obtain homogenous complete cement without K^+^, Na^+^, and Mg^2+^. TMAH was used to exclude the expansion contribution of ASR. Its molecular formula is (CH_3_)_4_NOH, and the relative molecular mass is 91.15. The alkalinity degree of TMAH is similar to NaOH and KOH, and it is a colorless deliquescent needle crystal with a melting point of 63 °C and a boiling point of 120 °C.

### 2.2. Preparation of Concrete Microbars and Rock Prisms

The modified concrete microbars (4 cm × 4 cm × 16 cm) were prepared with self-made cement and dolomitic rocks of 5–10 mm in grain size according to RILEM AAR-5 [22]. The weight ratio of aggregates to cement was fixed to 1/1 and the water-to-cement ratio was 0.32. In addition, 900 g of cement was used for each batch for three concrete bars. For each microbar formulation, aggregates and cement were mixed in a mixer (Type NJ-160A, Luheng Co., Ltd., Shanghai, China). All specimens with the mold were placed in a moist environment (RH = 98%) at 20 °C. After 24 h, the specimens were taken from the molds. Then, the initial length measurement was taken and the bars were transferred to containers filled with TMAH solutions. Rock prisms were prepared according to ASTM-C586 [24]. Rocks were modified by a cutting machine to obtain prism rocks with of 4 cm in length, 1 cm in width, and 1 cm in height. Nails were fixed at both ends of the rock prisms by glue and cement. Prepared rock prisms were cured in a 1-mol/L TMAH solution at 80 °C. After all specimens were cured to the corresponding age, the expansion rate test and calculations were performed, and the curing solution was changed every 14 days. The formulas for calculating the expansion ratio of concrete microbars and rock prisms are shown in Equations (3) and (4), respectively:(3)$$P_{t}=\frac{L_{t}-L_{0}}{145}\times 100\%$$
(4)$$P_{t}=\frac{L_{t}-L_{0}}{L_{0}-5}\times 100\%$$
where $P_{t}$ is the expansion rate after t days of maintenance, %; $L_{t}$ is the test piece length after t days of maintenance, mm; and $L_{0}$ is the initial length of the test piece, mm.

### 2.3. Testing and Characterization

X-ray diffraction (Smart Lab, Rigaku, Tokyo, Japan) analysis was used for the composition of the self-made cement without K^+^, Na^+^, and Mg^2+^ and the composition of dolomitic rocks. The length changes of all specimens prepared by different aggregates were measured at different ages and the expansion ratio of all the specimens were calculated by the abovementioned formula. Each length change value used was the mean value of three replicate specimens. The morphologies of aggregate grains enriched dolomite selected from the microbars cured in TMAH solution were also observed by SEM (JEOL, JEM-6510, Tokyo, Japan) coupled with EDS analysis. Orthogonal polarized light microscopy (DM750P, Leica, Germany) was also used to investigate the cracks as a result of ACR.

## 3. Results and Discussion

### 3.1. The Expansion of Concrete Microbars and Rock Prisms Cured in TMAH Solution

It can be seen from Figure 4a that the expansion rate of the concrete microbars cured in the 1-mol/L TMAH solution at 80 °C was slow before but increased after 28 days. At early curing ages, samples with slow expansion can be ascribed to the obvious shrinkage of cement without alkali, and the shrinkage offset some expansion originating from ACR. Cement shrinkage can be ascribed to the formation of hydration products with lower volume during the early ages. At the later age, the expansion rate of the concrete microbars was relatively stable and the expansion was obvious. This indicated that the contribution of the ACR to expansion was carried out later. The expansion characteristics of the microbars were similar to the results from López-Buendía [6].

From Figure 4b, combined with Table 1 and alkali activity analysis, the SiO_2_ content of these rocks was more than 9% (9–26%), and these rocks had both ACR and ASR activity or both ACR and potential ASR activity. It was found that the expansion rate of the rock prisms in the 1-mol/L TMAH solution at 80 °C increased with curing age. This proves that even in the case where the SiO_2_ content was high, since the TMAH solution did not react with SiO_2_, the alkali ions in the TMAH solution had an ACR with the dolomite and expand. The whole expansion of the rock prisms was not particularly big due to the TMAH solution ruling out the expansion coming from ASR. In the early stage, the expansion of the rock prisms was slow, and the expansion rate increased significantly after 72 days. Among them, the rock prism CX1 possessed the largest expansion. This is because the alkali solution entered the test pieces slowly, and there were relatively fewer alkali ions in the rock prisms at the early age. Therefore, the degree of ACR was lower than that at the later age.

Comparing Figure 4a and Figure 4b, it can be observed that at the same age, the expansions of concrete microbars were lower than the rock prisms made of the same rock. This is because the rock prism reacted with the alkali solution due to direct contact. However, the aggregates in the concrete microbar were covered by the cement, hindering the reaction between alkali and aggregate. In addition, the self-shrinkage of cement could offset some expansion that originated from ACR.

### 3.2. Crack Characteristics

Concrete microbars and rock prisms cured in TMAH solution were made into thin slices and observed by a polarizing microscope. Figure 5, Figure 6 and Figure 7 show the cracks of the concrete microbars at different ages cured in the 1-mol/L TMAH solution at 80 °C. It was found that expansion cracks occurred in the dolomite region. The cracks of the rock prisms are shown in Figure 8 and Figure 9. In Figure 5, Figure 6 and Figure 7 and Figure 9, the red arrow points to the crack and the yellow arrow points to the dolomite crystal. In Figure 8, the red arrow points to the crack, the yellow arrow points to the calcite region, and the green arrow points to the dolomite region.

As shown in Figure 5a,b, the dolomite and calcite enrichment areas of this rock were obviously decomposed. In addition, dolomite crystals were dispersed, and the crack was produced by ACR between dolomite crystals and alkali ions. In the calcite enrichment area, only this extended crack was generated by the ACR between the dolomite crystals and the alkali ions. Since the alkali solution first infiltrated from the edge of the rock, the reaction also proceeded from the edge of the rock. If the crack had been generated in the calcite enrichment area, the crack should have been gradually reduced in the process of extending to the outside (the dolomite enrichment area). However, the crack with the larger width was closer to the boundary between the dolomite and calcite enrichment areas, as shown in Figure 5b. It was obvious that this crack extended from the dolomite enrichment area to the calcite enrichment area. This crack was not caused by calcite. As shown in Figure 6a,b, dolomite crystals were embedded in a calcite matrix. A large number of dolomite crystals were distributed around the crack. The crack was only inside the rock and did not extend into the cement paste. Therefore, it can be inferred that the crack was an expansion crack generated inside the rock, and the expansion source was dolomite crystals. The crack shown in Figure 7 was generated inside the rock and extended into the cement paste. Different sizes of dolomite crystals were distributed at the starting position of the crack. It can be seen that those cracks, shown in Figure 7a,b, are expansion cracks, and the expansion source was dolomite crystals.

Figure 8 shows the comparison before and after erosion of the polished surface of the rock prisms using hydrochloric acid. Hydrochloric acid was used to facilitate the distinction between the dolomite and calcite regions of the rock prisms and helped in observing the position where the crack generated. In addition, it was used to investigate the characteristics of cracks only caused by ACR and the distribution of cracks. All cracks generated in the dolomite region indicated that the cracks resulted from ACR. Comparing Figure 8a and Figure 8b, it was found that after erosion of the polished surface of the rock prisms, some cracks on the surface disappeared. These cracks were eroded by hydrochloric acid. After erosion, the boundary between the calcite and dolomite regions on the surface of the rock prism was very obvious. The differences can be seen in color and height. The surface of the calcite region was lower than the dolomite region. The dolomite region was grayish white, and the calcite region was black. This color difference could also be observed on the polished surface of the rock prism before erosion. It can be observed that some cracks were parallel to the length direction of the rock prism and some cracks were perpendicular to the length direction before erosion. After erosion, it can be observed that those cracks were in the dolomite region.

It can be seen from Figure 9 that all of the dolomite crystals were in a mosaic distribution around the crack. In addition, when observing the crack of the rock prism, it was found that the closer to the central axis of the rock prism, the less that cracks were generated. Also, many cracks were found on the surface of the rock prism. This could have resulted from that the alkali solution starting from the rock surface when it infiltrated into the rock interior, so the migration of alkali ions into the rock also started from the rock surface, and the alkali–carbonate reaction first proceeded on the rock surface. This phenomenon could also be observed in the concrete microbars, as shown in Figure 6a and Figure 7a. In summary, both mosaic distribution or dispersion distribution of dolomite crystals could react with alkali and produce expansion cracks, and the expansion source was dolomite crystals.

### 3.3. Products Analysis

Based on characteristics analysis of the cracks caused by ACR, light microscopy was prepared from the concrete microbars and rock prisms cured in the TMAH solution. After being applied for 24 h in a clean wet towel, no ASR gel was observed under a stereo microscope. Katayama [2,5] investigated samples cured in NaOH by polarizing microscopy and concluded that alkali–carbonate reaction produced a myrmekitic texture, which was composed of spotted brucite and calcite within the reaction rim. However, to further investigate the reaction products of ACR, SEM-EDS analysis was carried out. As shown in Figure 10, the reacted JT1 aggregate with the dolomite enrichment area and cracks was selected to be observed by SEM-EDS analysis. Based on EDS analysis, positions 1–3 were calcite, brucite, and dolomite, respectively. Milanesi et al. [13] found brucite and calcite in the process of ACR according to X-ray diffraction analysis. Here, the reaction products of ACR were similar to the results of Milanesi et al. No ASR gel was found in the cracks as well. According to the SEM images, it can be seen that rod-like crystals were found around the dolomite crystals, and the rod-like crystals were brucite. Therefore, these products were considered to be brucite crystals resulting from ACR.

## 4. Conclusions

In this study, self-made cement and TMAH solution were used to avoid the effect of ASR on the expansion of the specimen, and the expansion and cracks only caused by ACR were investigated. From the physical measurement and microstructural analysis, the following conclusions can be obtained.

In the dolomite enrichment area, both dispersion and mosaic distributions of dolomite crystals could react with alkali ions in the TMAH solution. In the early curing age, the expansion of concrete microbars was low due to self-shrinkage of the cement. The degree of ACR increased continuously, with the alkali solution gradually increased at the later stage and the expansion rate of the specimen gradually increased, leading to expansion and cracking of the specimen. Therefore, dolomite reacted with TMAH and caused the concrete microbars and rock prisms to expand, and the contribution of the ACR to expansion was carried out later.

Based on microscopic analysis, expansion cracks were generated in the dolomite enrichment area, and dolomite crystals with different particle sizes were distributed at the crack origin as well as around the crack. These dolomite crystals were the expansion source. The expansion cracks either extended inside the rock or into the cement phase.

The SEM-EDS analysis indicated that rod-like brucite crystals were formed in the process of ACR. The reaction products of ACR, including brucite and calcite, were distributed around dolomite crystals. No ASR gels were found in concrete microbars and rock prisms. Namely, the ASR did not occur in the entire reaction system.

## Figures and Tables

**Figure 1 materials-12-01228-f001:**
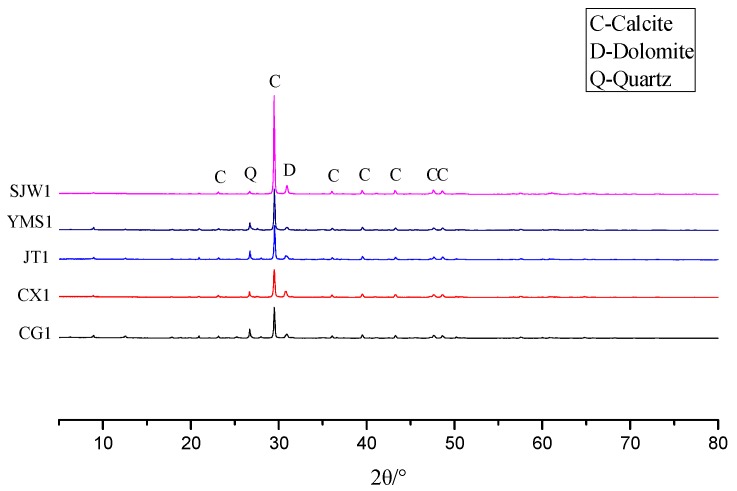
The XRD pattern of different rocks.

**Figure 2 materials-12-01228-f002:**
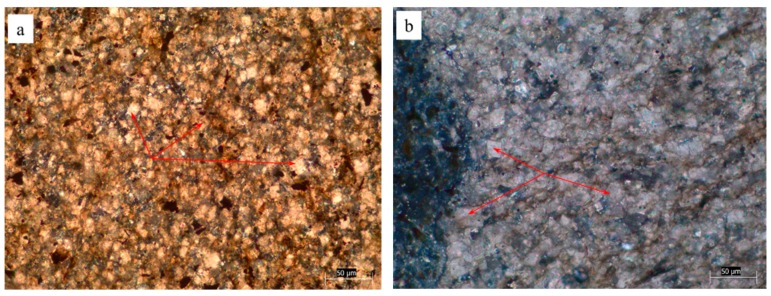
The distribution of dolomite crystals inside YMS1 and CG1 rocks: (**a**) dispersive distribution, (**b**) mosaic distribution.

**Figure 3 materials-12-01228-f003:**
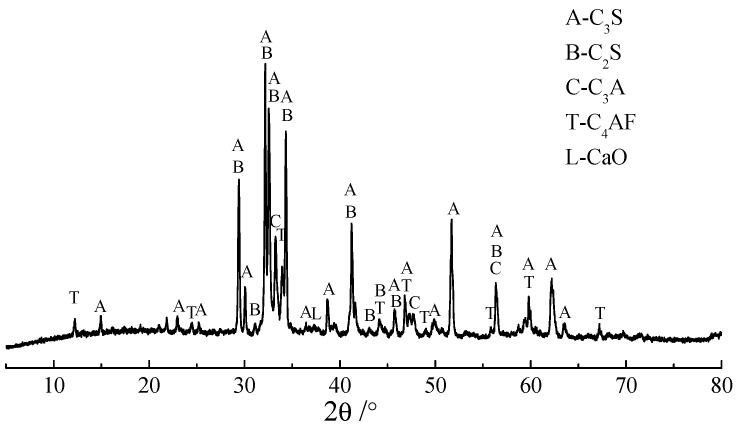
The XRD pattern of self-made cement clinker.

**Figure 4 materials-12-01228-f004:**
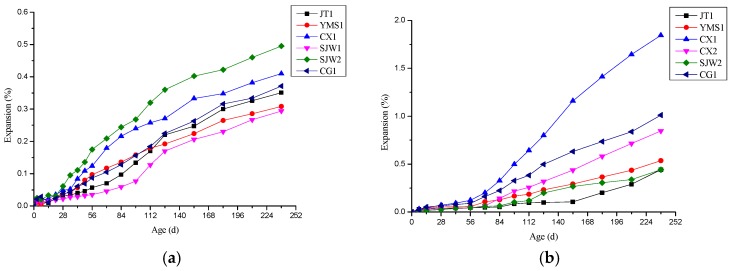
Expansion rate curve of specimens cured in tetramethylammonium hydroxide (TMAH) solution: (**a**) concrete microbars, (**b**) rock prisms.

**Figure 5 materials-12-01228-f005:**
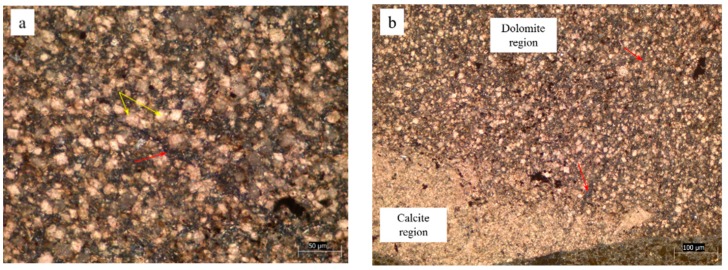
Expansion cracks in concrete microbar of YMS1 cured in TMAH solution for 56 days: (**a**) crack in dolomite region; (**b**) crack in calcite region.

**Figure 6 materials-12-01228-f006:**
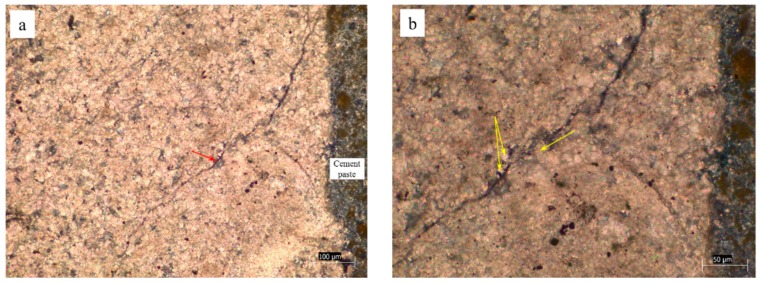
Expansion cracks in concrete microbar of YMS1 cured in TMAH solution for 98 days: (**a**) crack in dolomite region; (**b**) dolomite around the crack.

**Figure 7 materials-12-01228-f007:**
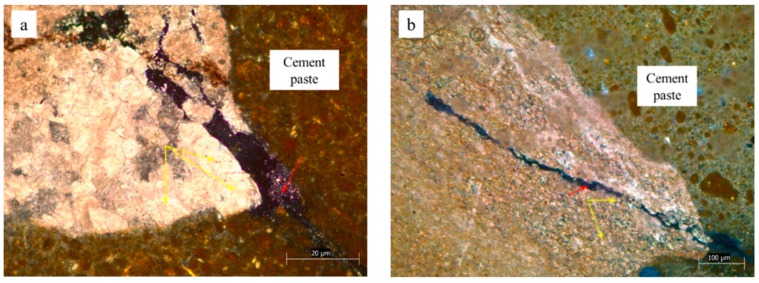
Expansion cracks in concrete microbar cured in TMAH solution for 154 days: (**a**) CG1, (**b**) JT1.

**Figure 8 materials-12-01228-f008:**
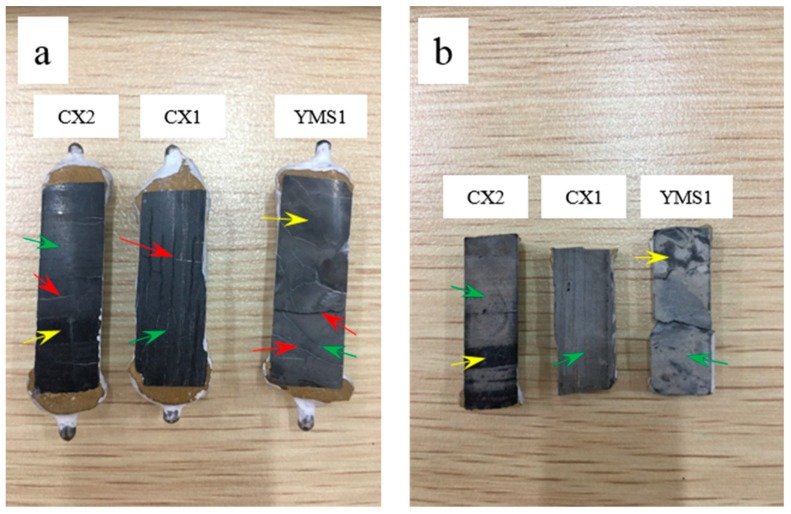
Rock prisms cracks and erosion surface cured in TMAH solution for 154 days: (**a**) before erosion, (**b**) after erosion.

**Figure 9 materials-12-01228-f009:**
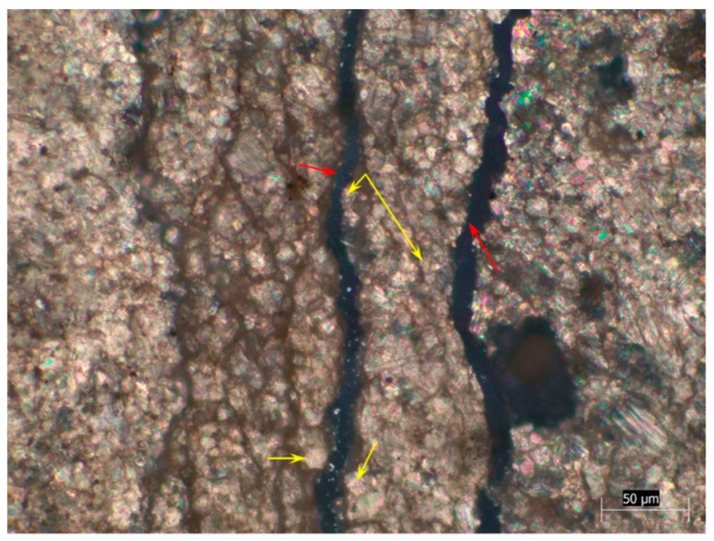
Rock prism cracks of CX2 cured in TMAH solution for 154 days.

**Figure 10 materials-12-01228-f010:**
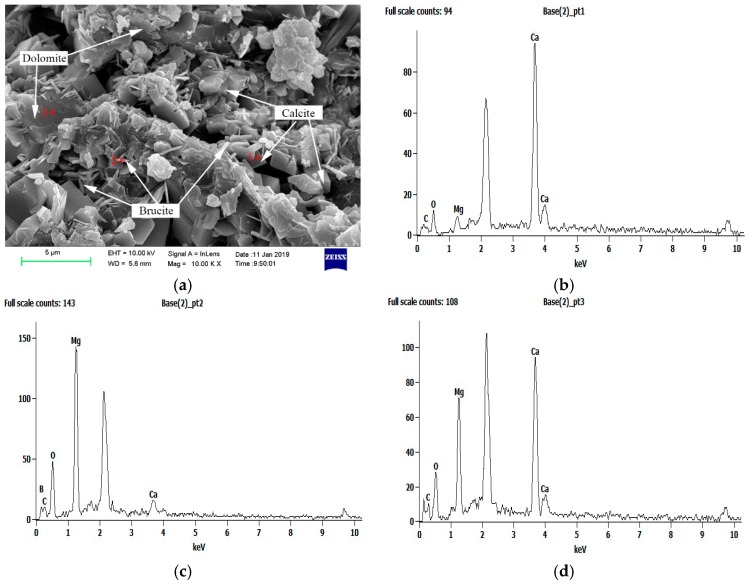
Scanning electron microscopy and energy dispersive spectrometry (SEM-EDS) analysis: (**a**) SEM image of reacted JT1 aggregate; (**b**–**d**) EDS images of calcite, brucite, and dolomite, respectively.

**Table 1 materials-12-01228-t001:** Rock chemical compositions.

Rock		Composition (wt.%)					
		LOI	SiO_2_	Fe_2_O_3_	Al_2_O_3_	CaO	MgO
1	CG1	29.26	22.47	2.59	6.59	31.70	4.29
2	JT1	28.60	23.36	2.84	7.25	31.15	3.58
3	YMS1	26.48	25.02	2.87	7.82	29.50	3.09
4	CX1	33.94	15.21	1.81	4.30	38.26	2.92
5	CX2	35.41	12.95	1.94	3.96	40.63	3.12
6	SJW1	37.91	9.10	1.16	2.96	42.75	4.29
7	SJW2	36.72	11.69	1.43	3.82	39.27	4.58

**Table 2 materials-12-01228-t002:** Raw material composition of cement clinker without alkali (wt %).

CaCO_3_	SiO_2_	Al_2_O_3_	Fe_3_O_4_	Total
78.18	14.03	4.40	3.39	100

**Table 3 materials-12-01228-t003:** Content of minerals in self-made cement clinker (%).

C_3_S	C_2_S	C_4_AF	C_3_A	f-CaO	f-MgO
64.0	11.9	10.9	13.2	0.1	0

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
