# Peer review of "The Expansion Cracks of Dolomitic Aggregates Cured in TMAH Solution Caused by Alkali–Carbonate Reaction"

_materials, 2019, doi:10.3390/ma12081228_

Reviewer 1 Report

The reviewed manuscript deals with an interesting problem of alkali-aggregate reaction (AAR) testing that allows the distinction between its causes (be it alkali-silica reaction (ASR) or alkali-carbonate reaction (ACR)). According to the Authors, the TMAH solution produces ACR without causing ASR. However, the reasons for this effect are not explained. The presented test results can be considered valuable, but they require adequate description.

 1.      L28-29: “The ACR is also called alkali-dolomite reaction (ADR) or dedolomitization” - by whom? Give references. ACR is a broader concept than dedolomitization, see Katayama [4]. ADR is also an unusual abbreviation.

 2.      L53: not “slurry” but “cement paste” or “cement matrix”.

 3.      Section 1. “Introduction”: I would advise to broaden the description of ACR theory. For example, pay attention to the role of clay minerals or dolomite lithology etc. I suggest the Authors pay attention to these papers:

- LĂłpez-Buendia A.M., Climent V., VerdĂş P., Lithological influence of aggregate in the alkali-carbonate reaction, Cement and Concrete Research 36 (2006)
1490-1500.

- Tong, L., Tang, M., Correlation between reaction and expansion of alkali-carbonate reaction, Cement and Concrete research 25 (1995) 470-476.

- Pagano M.A., Candy P.D., A chemical approach to the problem of alkali-reactive carbonate aggregates, Cement and Concrete Research 12 (1982) 1-12.

 4.      What is the explanation for TMAH not causing ASR only ADR? - it should be explained in the "Introduction".

 5.      Section 2.1: The rocks used should be described better. What are the differences and similarities between them?  I suggest the Authors present a petrographic analysis (The XRD analysis can not fully replace the observation in optical microscope).

 6.      Section 2.1: The sodium and potassium play an important role in the Portland cement clinker sintering. Due to the modification of this process, providing the cement phase composition would be advisible.

 7.      L115-116: “X-ray diffraction (Smart Lab, Rigaku, Tokyo, Japan) analysis was used for the composition of self-made cement without K+, Na+ and Mg2+ 116, and the mineralogical detection of dolomitic rocks.” – these results are not included in this paper.

 8.      Section 2.2: The concrete micro-bars tests should be described better. Which standards were used? The particle composition of aggregate must be presented more accurately.

 9.      Fig. 2a. and b.: I recommend using the same designations for the same rocks (as for JT1 and CG1).

 10.  L133: “From Figure 2 (b), combined with chemical analysis and alkali activity analysis” - there are no chemical analysis results in this figure.

 11.  L187-L196: What do these observations have to do with ACR? - explain. In section 2.3. these tests should be described better. Why was hydrochloric acid used?

 12.  L216-217: The presence of calcite and brucite cannot be confirmed based on Fig 8a.  For this purpose, the results of spot analysis (EDS) and microstructure images taken at higher magnification could be useful.  I would also suggest the Authors reduce image contrast.

 13.  The results should be described against available literature.  Section 3 contains no citations. This must be improved.

 14.  Improve spelling, e.g:

- L10 “In” is bold

- L11. “ tetramethy1 ammonium hydroxide” – should be Tetramethylammonium hydroxide

- L152 “Crock prisms”?

 Author Response

All the comments are very valuable and critic to improve our submitted manuscript. Based on the comments we received, the manuscript has been carefully revised and the revisions are highlighted in red. We would like to express our deep gratitude to you for the comments in our paper.

Reviewer 2 Report

Line 59 - Explain why TMAH reacts with dolomite in carbonate rock and expand, but does not react with ASR active components.

Provide table for mixture proportion. 

What was water-to-cement ratio?

Authors to describe a summary of ASTM-C586.

Line 118 - Ratio or Radio?

Authors to provide more literature review.

Author Response

All the comments are very valuable and critic to improve our submitted manuscript. Based on the comments we received, the manuscript has been carefully revised and the revisions are highlighted in red. We would like to express our deep gratitude to you for the comments in our paper.

Round  2

Reviewer 1 Report

The manuscript is generally well prepared. However, there are a few items that require improvements:

 1.       Fig. 2, 5-7 and 9: The scale should be added;

2.       Equations 1 and 2 are of poor quality (letters are blurred);

3.       Fig. 10 should be improved. Fig 10a is not readable enough - it is difficult to match the EDS           analysis and the spot in Fig. 10a or read the magnification used in Fig. 10a. The caption                under Fig 10 could be more accurate.

Author Response

The manuscript has been carefully revised and the revisions are highlighted in blue.

Reviewer 2 Report

This paper can be published in its present format.

Author Response

Thank you very much for your comment.